Leveraging deep learning for toxic comment detection in cursive languages

Shahid Muhammad 1
Umair Muhammad 1
Iqbal Muhammad Amjad 1
Rashid Muhammad 2
Akram Sheeraz 3 4 5
Zubair Muhammad muhammadzubair@ucp.edu.pk engr.zubairfarooqi@gmail.com 1
1 Faculty of Information Technology and Computer Science, University of Central Punjab , Lahore , Pakistan
2 Department of Computer Science, National University of Technology , Islamabad , Pakistan
3 Information Systems Department, College of Computer and Information Sciences, Imam Mohammad Ibn Saud Islamic University , Riyadh , Saudi Arabia
4 Faculty of Computer Science and Information Technology, The Superior University , Lahore , Pakistan
5 Intelligent Data Visual Computing Research (IDVCR), The Superior University , Lahore , Pakistan
Alatas Bilal
Electronic publication date: 2024 Dec 13
Publication date: 2024
Volume: 10
Electronic Location ID: e2486
Received 2023 Jul 12; Accepted 2024 Oct 16
Copyright: ©2024 Bhatti et al.
Copyright year: 2024
Copyright holder: Bhatti et al.
License: This is an open access article distributed under the terms of the Creative Commons Attribution License, which permits unrestricted use, distribution, reproduction and adaptation in any medium and for any purpose provided that it is properly attributed. For attribution, the original author(s), title, publication source (PeerJ Computer Science) and either DOI or URL of the article must be cited.
License URL: https://creativecommons.org/licenses/by/4.0/

Keywords: Toxic comments, Cursive languages, Corpus, Deep learning, Classification

Funding: The authors received no funding for this work.

==============================
Social media platforms enable individuals to publicly express opinions, support, and criticism. Influencers can launch campaigns to promote ideas. Most people can now share their views and feelings through visual or textual comments, which can range from appreciation to hate speech, potentially inciting societal violence and hatred. Detecting these noxious comments and thoughts is critical to protecting our communities from their negative social, psychological, and political impact. Although Urdu (a low-resource language) is one of the most popular Asian languages around the globe, a standard tool does not exist to detect toxic comments posted in this language. Tokenization and then categorizing cursive text is challenging due to its complex nature, especially when dealing with toxic comments, which are often ungrammatical and very brief. This study proposes a novel model to identify salient features in Urdu sentences. It uses transformers to identify and flag toxic comments using deep learning binary classification of the text. Statistically, the proposed fine-tuned model outperforms the existing ones by achieving a precision of 88.38%. Among the models evaluated, bidirectional encoder representations from transformers (BERT) demonstrated superior performance with an accuracy 85.45%, precision 85.71%, recall 85.45%, F1 score 85.41%, and a Cohen Kappa 70.83% on the full feature set. Conversely, GPT-2 was identified as the lowest-performing model. The outcomes of this research represent a noteworthy advancement in the broader endeavor to improve and optimize content moderation mechanisms across diverse languages and platforms.

Background and Significance of the Study

A huge amount of data is created and circulated on the internet daily. This data often takes the form of comments posted on various forums through text-based input forms and is accessible to all users. Sharing information and expressing opinions in this way contributes to social awareness and enhances the quality of information available. However, publicly shared comments may also contain toxic content that can damage an individual’s reputation or even constitute online harassment. This issue has mobilized the research community to develop statistical models to predict and prevent such situations. It is critical to classify these comments accurately in order to address the problem effectively.

There is currently no categorization technique to identify hostile words or phrases in Urdu. However, the Arabic language is closely related to Urdu, particularly in terms of writing scripts. For Arabic, studies such as Mukund & Srihari (2012), Dehghani (2024) have worked on sentiment classification using convolutional neural network (CNN), achieving a success rate of 87%. Therefore, a classification model is needed to classify offensive texts written in Urdu.

The amount of text data is constantly increasing, creating the need to develop new machine learning methods to address the problem effectively. Improvements in computer hardware, the availability of cloud computing paradigms, and the introduction of big data techniques have resulted in the development of methods that provide better performance. For text classification, in particular, several neural network models (Georgakopoulos et al., 2018a; Ibrahim, Torki & El-Makky, 2018; Saeed, Shahzad & Kamiran, 2018) have been proposed for text analysis, emphasizing the structure of the text within documents.

Participating in online discussions can sometimes be troubling due to provocative conversations, which may cause people to refrain from using online forums. This research aims to facilitate effective exchanges and limit user comments on some forums. This classification may lead to closed or limited conversations that contain toxic comments. Social networks and news websites allow their users to post comments, and they often face challenges caused by unethical comments in the form of legal objections. The worst part is that such activities may lead some segments of society to spread hate or false propaganda, causing panic or gaining political advantage. As a result, the situation can create a confrontational environment that discourages healthy debate and promotes an abusive culture. Any user facing such a situation may never visit that forum again, resulting in fewer users for that particular web resource.

There are many parallels between the writing systems of cursive languages like Urdu, Arabic, and Persian, including a right-to-left direction and a flowing, continuous script. This table compares phrases commonly used in these languages with their English equivalents and provides a severity rating based on their potential offensiveness. The adjectives used in this comparison range from negative (such as “rude”, “disrespectful”, “selfish”, “obscene”, and “low-quality”) to neutral (such as “clean”, “friendly”). Despite being spoken in different languages, there is significant overlap in these terms. For example, the English word “rude” is translated from Urdu, Arabic, and Persian terms, all of which have a severity level of 3. It is intriguing to observe how comparable neutral or offensive concepts in one language may be in another. For example, while friendliness and cleanliness are valued in many cultures, rudeness and disrespect are not. These similarities highlight the shared linguistic and cultural roots of these cursive languages. However, it is also critical to recognize the distinctive nuances and connotations of each language’s words. These subtle shades of meaning may not be fully captured when words are translated into English, illustrating the depth and richness of these languages.

Understanding the similarities and differences between these cursive languages in terms of their potential negative and neutral words can be aided by Fig. 1. It provides a glimpse into the diverse and fascinating world of cursive languages, highlighting both their commonalities and distinctions.

Figure 1 Examples of toxic and non-toxic words in Urdu, Arabic, and Persian with severity levels based on language expert opinion.

Offensive comment categorization falls into the general category of text classification, where a sample comment or a list of comments is classified using a trained model to determine whether it is toxic or neutral. In this research, only a binary categorization of comments in Urdu has been made. The above-quoted text example can help us understand this.

Another example is a search engine such as Google, when you type in a search to do so, it shows an inappropriate list that may be inappropriate for a particular nation or religion. An example of inappropriate text shown during a search query can be seen in Fig. 2. In this example, the name of a political character was explored on Google, so Google received some subtle suggestions. In the same way, when searching for religion on Yahoo, the response to the query contains some inappropriate suggestions that might upset a large number of people.

Figure 2 An example of inappropriate text shown during a search query.

The rest of the article is distributed into the Literature Review, Corpus Generation, Data Pre-processing, Machine Learning Model Selection, Simulation and Results, and Conclusion sections.

Novel contributions and innovations

The present study presents a significant advancement and a novel approach to cursive language text categorization and characteristic extraction. Our work goes beyond conventional methods and tackles the unique challenges these languages pose. We specifically focus on Urdu, but our approach is applicable to other cursive languages such as Arabic and Persian. Initially, we present an innovative method for extracting features from cursive language strings that employs unsupervised learning techniques. The methodology used in this study involves the implementation of bidirectional encoder representations from transformers (BERT) for several essential tasks, such as tokenization and the identification of lemma words. Given the complexity of cursive scripts and their inherent linguistic nuances, this represents a significant advancement over traditional methods.

Secondly, we pioneered the application of memorization algorithms for text categorization in cursive languages. This approach provides an effective solution for dealing with the high dimensionality and complexity of cursive language data.

Current State of the Art

To date, several relevant research studies have been conducted in the sentiment analysis domain that deals with toxic comments. However, the explored work is classified based on relevant approaches or techniques, not semantic analysis.

Recently, Babakov, Logacheva & Panchenko (2024) and Li et al. (2024) used GPT and BERT to reduce biases in the categorization of normal and toxic text. The efficacy of various multilingual language models (Pamungkas et al., 2023) in identifying instances of hate speech in code-mixed Indonesian social media data, with a particular focus on Javanese-Indonesian and Sundanese-Indonesian languages. Three distinct models were evaluated using three existing multilingual language models, namely LASER, Multilingual FastText (M-FastText), and Multilingual BERT (M-BERT), and their efficacy was juxtaposed with monolingual Indonesian language models. The training of the models was performed on a dataset partitioned into a 70% training set and a 30% test set, with minimal preprocessing procedures applied. The findings indicate that the performance of the multilingual models was not superior to that of the monolingual Indonesian models. The results indicate that the Indonesian BERT model exhibited superior performance compared to the multilingual BERT model across the Javanese-Indonesian and Sundanese-Indonesian datasets. The researchers attributed this phenomenon to the prevalence of Indonesian lexemes in the code-mixed dataset, which replaced the usage of local language lexemes. The researchers discovered that deep learning models exhibited superior performance compared to conventional models in all datasets and language configurations. The research findings indicate that the implementation of multilingual models did not yield any significant performance improvement. However, the study suggests that alternative approaches, such as transfer-of-knowledge approaches, may hold promise for further improvement.

Albert (2022) and Berrimi (2024) offers a methodical examination of the existing literature pertaining to the identification of hate speech across different domains. To facilitate the analysis, the PRISMA framework was employed as a guiding tool. The authors identified several significant challenges in the field, such as non-standard grammar and vocabulary, insufficient data, representation bias, implicit expressions, various forms of hatred, overlooked knowledge in the target domain, and gender targeting. The authors also conducted an assessment of extant datasets and identified six that hold potential for employment in forthcoming cross-domain adaptation endeavours. The analysis demonstrated that contemporary deep learning methodologies have augmented the emphasis on cross-domain hate speech. However, there are still constraints and obstacles that need to be surmounted. A plausible criticism could be that the efficacy of these models may be limited for languages with intricate scripts or cursive writing systems, as the investigation did not encompass such languages. The study’s authors concluded that further investigation is required to formulate versatile and adaptable frameworks for the identification of hate speech.

Khan, Shahzad & Malik (2021) and Dewani et al. (2023) made a commendable effort to bridge this gap. They developed a Hate Speech Roman Urdu 2020 corpus. Additionally, they compared the effectiveness of multiple supervised learning techniques for hate speech detection in this language. However, the research’s primary focus on Roman Urdu, which is a more formal and less widely used script, potentially limits its applicability in real-world scenarios. The author identified a notable inadequacy in the system’s inability to tackle the problem of identifying hate speech in handwritten Urdu script, which is often characterized by its informal and unstructured nature and is commonly employed in everyday conversations.

Saleem et al. (2017) used various algorithms to create language models for detecting hate speech. Their proposed model took input from a user and split the text based on predefined community labels. They used naive Bayes (NB), support vector machine (SVM), and logistic regression (LR) with unigram and TFIDF weights on the given dataset. This research work was only used for the categorization of English language commentaries. Djuric et al. (2015b) developed an algorithm to detect only obscene text comments from social media applications by using SVM and naive Bayes classifiers.

Ranasinghe & Zampieri (2021) and Aken et al. (2018) compared different deep learning and shallow approaches on a large dataset with comments. Further, they validated findings on another dataset. Then they performed error analysis by testing on the dataset and validating one another dataset and identified the missing paradigmatic context and inconsistent dataset labels.

Google created the Perspective API (https://www.perspectiveapi.com), recognizing the sensitivity of the impact of toxic comments on society. The goal is to make spam comment detection and removal easier on online forums. To achieve this, Google enabled people to rank these comments to determine their level of toxicity. Subsequently, new comments were evaluated based on the same rating system. The limitation of the initial version of the API is that it provides support for the English language only. Hosseini et al. (2017) devised a way to deceive the Google API. The authors showed that the toxic ranking could be manipulated by attacking the toxic score levels. The proposed system deceived the API, causing harmful comments to be considered non-toxic.

Wulczyn, Thain & Dixon (2016) developed a methodology for Wikipedia that involved labeling 100k comments by humans and 63 million automatically labeled by computers to explore questions about trolling comments. They developed a model to create large-scale data about personal harassment in online forums. After obtaining the results of crowdsourcing toxic comment rankings and machine-labeled rankings, they scaled the identification to the whole corpus. They introduced some bias based on human judgment of comments into the training labels to compare machine models to manual annotations.

Parekh & Patel (2017) reviewed several methodologies (Djuric et al., 2015a; Kansara & Shekokar, 2015; Yadav & Manwatkar, 2015) that had previously been applied to detect toxic comments on social networking sites. She categorized these techniques and highlighted their weaknesses.

Nobata et al. (2016) developed a learning method to identify abusive text in online user posts from a domain that uses N-gram, linguistic, and syntactic features. Khatri et al. (2018) introduced a two-stage semi-supervised model for the automated detection of violent text from publicly available resources. They used various data collection methods, first by using blacklist ranking sensitivity of comments followed by a sampling of random text, and second by a semi-supervised approach using online forums to create a text collection. The authors performed a comparative analysis of all models on Twitter and Wikipedia-sensitive comments. Their model performed better than the previously set baseline score, improving from 75% accuracy to 95.5%.

Zhang & Luo (2018) used a deep neural network to extract features that are particularly important for detecting toxicity-related semantics. They evaluated their method on a large dataset of hate speech and improved baseline results by 5% in macro-average F1. Some researchers used mixed language to confront the challenge. In Srivastava, Khurana & Tewari (2018), the authors used the TRAC shared task dataset consisting of Hindi and English comments. They further experimented on the Kaggle toxic comment dataset, achieving an accuracy of 98.46% (Srivastava, Khurana & Tewari, 2018).

Gaydhani et al. (2018) developed an approach to automatically classify tweets on Twitter into three categories: hateful, offensive, and clean. Using the Twitter dataset, they performed experiments using n-grams as features and passed their Term Frequency-Inverse Document Frequency (TFIDF) values to multiple machine learning models. Comparative models were analyzed using several values of n in n-grams and TFIDF normalization methods. After tuning the model to give the best results, they achieved 95.6% accuracy upon evaluating it on test data. They created a module that served as an intermediary between users and Twitter.

Jahan & Oussala (2023) conducted a comprehensive review of deep learning models applied to abusive text classification. Their analysis revealed that 12–14% of the studies utilized CNN as standalone models, 8–9% employed hybrid models combining CNN with LSTM, and 2% incorporated features from BERT with CNN. The study highlighted the variability in performance across languages, with the model achieving an F1 score of 93 for English but dropping significantly to 67 for less-resourced languages such as Danish.

Georgakopoulos et al. (2018a) studied CNN-based approaches that have recently been used for toxic comments. They claimed to have sufficient evidence that the performance of CNN was better than any other model used for toxicity detection. The results were further encouraging for other CNN-based work in text classification. Georgakopoulos et al. (2018b) used a large amount of text collected during Kaggle’s competition on Wikipedia’s talk edits. They used a CNN against a previously used bag-of-words approach and found it very effective for text categorization. The results showed that CNN improved toxic comment classification.

Gambäck & Sikdar (2017) used CNN to classify hate speech. They labeled all tweets with labels such as racism, sexism, both, or neutral comments. CNN was used to train on 4-gram word-based semantics created using word2vec. Randomly generated vectors consisting of words were coupled with n-grams and evaluated using 10-fold cross-validation, which used a word2vec model with an accuracy of 78.3%. Meyer (2018) developed a deep learning-based architecture to detect hate comments. They combined two models, CNN and LSTM. The authors used character and word n-grams for CNN and LSTM, and the results of those models were combined for classification. This two-pronged model, which used embeddings of both CNN and LSTM, produced better results than baseline results. The result was better when a two-layered CNN was used. However, the use of multi-layering on LSTMs reduced performance.

Vazhentsev et al. (2023) explored advanced deep learning models, including BERT and ELECTRA, on the PARADETOX dataset, a collection of English-language tweets annotated for toxicity. The authors proposed a novel uncertainty estimation framework that combines epistemic and aleatoric uncertainty estimation methods, enabling robust classification of toxic content. This work emphasizes binary classification for English tweets, providing a strong foundation for understanding model behavior under uncertain conditions.

Mubarak, Darwish & Magdy (2017) used the unigram word model trained on Arabic text to classify abusive words from the comments dataset. This was achieved by using a lookup table of offensive words. The authors evaluated their methods on multiple lists, i.e., unigram and bigram, for binary classification. The result was acceptable when using the unigram model only.

In conclusion, as mentioned earlier, the research projects were carried out in English only as shown in literature review summary in Table 1. Toxic comments in English have been significantly reduced, but the same cannot be said for cursive languages like Urdu. Although widely spoken in Asia, Urdu needs to be addressed when developing tools to detect toxic comments. This has become a critical issue as more people use Urdu for online communication on social media platforms. Effective tools that can identify and moderate toxic comments in Urdu are needed. This study proposes a novel model that uses deep learning and transformers to identify important features in Urdu sentences, making it easier to detect and flag toxic comments. The following are the novel contributions of the study.

Table 1 Summary of literature review on toxic comment detection.

Study	Approach/Model	Dataset/Language	Key findings	
Babakov, Logacheva & Panchenko (2024); Li et al. (2024)	GPT, BERT	Multilingual, Javanese-Indonesian, Sundanese-Indonesian	BERT outperformed multilingual models; deep learning models superior to conventional models.	
Albert (2022); Berrimi (2024)	PRISMA framework	Cross-domain datasets	Identified challenges in hate speech detection; need for versatile frameworks.	
Khan, Shahzad & Malik (2021); Dewani et al. (2023)	Supervised learning	Roman Urdu	Developed corpus; highlighted limitations in handwritten Urdu detection.	
Saleem et al. (2017)	NB, SVM, LR	English	Focused on English comment categorization.	
Ranasinghe & Zampieri (2021); Aken et al. (2018)	Deep learning	Large datasets	Validated findings across datasets; identified labeling inconsistencies.	
Hosseini et al. (2017)	Google Perspective API	English	Demonstrated API manipulation vulnerabilities.	
Wulczyn, Thain & Dixon (2016)	Crowdsourcing, machine labeling	Wikipedia	Developed large-scale harassment data; introduced human judgment bias.	
Parekh & Patel (2017)	Various methodologies	Social networks	Categorized techniques; highlighted weaknesses.	
Nobata et al. (2016)	N-gram, linguistic features	Online posts	Improved accuracy from 75% to 95.5%.	
Zhang & Luo (2018)	DNN	Hate speech dataset	Improved baseline results by 5%.	
Gaydhani et al. (2018)	ML models with TFIDF	Twitter	Achieved 95.6% accuracy.	
Jahan & Oussala (2023)	CNN	Twitter	Achieved 91% accuracy; misclassified some tweets.	
Georgakopoulos et al. (2018a)	CNN	Wikipedia	CNN outperformed other models.	
Gambäck & Sikdar (2017)	CNN, word2vec	Twitter	Achieved 78.3% accuracy.	
Meyer (2018)	CNN, LSTM	Mixed datasets	Combined model improved results.	
Vazhentsev et al. (2023)	RNN with attenuation	GAZZETTA, MULTITOX	Improved RNN performance with attenuation.	
Mubarak, Darwish & Magdy (2017)	Unigram model	Arabic	Effective for binary classification.	

• Creation of a new dataset specifically for Urdu language toxic comment detection.

• Development of an effective tokenization process for handling cursive text.

• Fine-tuning of a model using deep learning and transformers for toxic text classification in Urdu.

Corpus Development Methodology

This section describes how the proposed toxic comments analysis corpora are created for Urdu. The steps include data collection, cleaning, preprocessing of the data, and associated annotations.

Data collection procedure

The dataset employed in this study was systematically collected using a methodical and structured approach to ensure an inclusive and representative sample of both non-toxic and toxic remarks. The initial step involved developing a Google form to facilitate and organize the submission of comments from participants.

This document was widely circulated among the undergraduate population of the university. The intention was to elicit multiple responses from each participant, spanning a spectrum of emotions from impartial to detrimental. This methodology facilitated the collection of a diverse array of remarks indicative of the wide range of human discourse.

Following the collection of form submissions, a meticulous review process was instituted to determine the genuineness and relevance of the data. A group of annotators systematically reviewed each submission, carefully analyzing the comments to assess their accuracy and relevance. A dual validation process was implemented to ensure that the dataset used in the study was not only voluminous but also relevant and of high quality, thus aligning with the research objectives.

After the conclusion of the data collection and validation procedures, the data was further refined through a sequence of preprocessing measures. These procedures included the normalization of textual data, the elimination of extraneous components, and the classification of the remarks. The compiled CSV file of the final dataset is a robust and comprehensive collection of remarks, ready for thorough examination and investigation.

The meticulous and methodical approach to collecting and organizing data not only yielded a comprehensive dataset for our current study but also established a standard for future research in this field. It highlights the importance of rigorous data collection and verification procedures.

Dataset pre-processing

The manual dataset curation necessitates a rigorous preprocessing stage. This involves a series of operations to clean and prepare the data for subsequent statistical analysis, as described in Fig. 3. Given the diverse nature of raw text data, cleaning involves several crucial steps. These steps ensure that irrelevant content does not significantly influence the results of language processing and classification tasks. The following are the steps taken for the preprocessing of the dataset.

Figure 3 Comprehensive data collection and preprocessing workflow for cursive language toxic comment analysis.

• Character normalization: Accented characters, incorrectly typed characters, and dialects in Urdu or similar languages within the comments that do not contribute to the meaning were normalized. This includes normalizing similar characters with different ASCII codes for uniformity.

• Tag and metadata removal: Textual data may contain various HTML tags, hashtags, and other metadata that do not contribute semantically to the comment’s content. These were programmatically removed during the cleaning process and further verified manually.

• Emoji and symbol removal: In modern digital communication, emojis and symbols are prevalent. However, these did not contribute to the textual data processing and were removed from the dataset.

• Punctuation and special character handling: Given that Urdu is written in the UTF-8 encoding scheme, which differs from ANSI, certain punctuation marks and special characters were converted into their encoding numeric value during data collection. For instance, double quotation marks (”) were replaced with their corresponding HTML code.

Upon completion of the initial cleanup phase, a comprehensive list of words was generated from all comments. This list was meticulously reviewed once again to remove any unnecessary residual content. Following this procedure, the UTOXS21 dataset was effectively cleaned of all extraneous elements, producing a high-quality, noise-free resource for subsequent analysis.

Dataset description

To sum up, our dataset UTOXS21 consisted of 17,078 short-length comments, of which 51.6% were nontoxic comments, and the remaining were manually marked as toxic comments. Characteristics of the dataset are shown in Table 2. The dataset consists of 17k Urdu language short text labeled comments (both toxic and nontoxic) on topics such as politics, sectarianism, identity, hate, and obscenity.

Table 2 Statistics summary of toxic and non-toxic comments.

Dataset statistics summary	
Statistic	Toxic	Non-toxic	
Total comments	8,259	8,819	
Total words	458,049	545,574	
Vocabulary size	15,323	18,667	
Avg. words in each comment	55	62	
Total characters in each comment	2,070,360	2,487,417	
Average characters in each comment	251	282	
Unique characters	85	104	

These selected comments conveyed either some toxicity or were nontoxic. To extract valuable information from the collected text, a month-long annotation process was performed by native Urdu experts. Two experts from the Urdu-speaking community reviewed the labels of each comment to ensure accuracy. Some comments were even hard to recognize by humans, so a third opinion was used to finalize the label.

The dataset distribution, with approximately 51.6% classified as nontoxic and the remaining comments identified as toxic, as shown in Fig. 4. The comments cover various subjects, such as politics, sectarianism, identity, hate, and obscenity. Native Urdu-speaking experts thoroughly reviewed and labeled each comment, which took a month to complete. To ensure accuracy, two experts independently reviewed each comment’s label, with a third opinion considered for any ambiguous comments. The following are the major distribution details of the dataset instances.

Figure 4 Class-wise dataset distribution of toxic and non toxic comments.

Dataset characteristics:

• Number of instances: 17,078 (8,259 of which were classified as toxic)

• Number of attributes: String attribute (vector of words) and the class

• Class distribution: 48.4% toxic and 51.6% nontoxic

Data structuring and preparation process

The experimental dataset is stored in a directory as plain text files. Each document is a UTF-8 format Urdu text with short comments. These file names contain class-label information. To avoid bias in the data, both toxic and nontoxic files are distributed as 48.4% toxic and 51.6% nontoxic. This corpus is pre-processed and contains clean text.

Specific linguistic considerations for Urdu

In contrast to short texts, analyzing grammar is easier in longer texts. Longer texts provide better meaning. Short texts often use informal vocabulary with spelling and grammatical errors. This is because proper grammar analysis methods work poorly in short texts. Many words in Urdu can have different meanings. In this situation, it becomes important to know the context of the solutions. Some of these examples are given in Fig. 5. Urdu is written using a derivation of the Persian script based on the Arabic alphabet. The following are some of the critical challenges in the Urdu language. There are no small and capital characters in the Urdu alphabet. Therefore, it is difficult to recognize the start of sentences and nouns. This makes it difficult to tag parts of speech. Urdu writing is bidirectional, with alphabets written from right to left and numbers written from left to right. It is a cursive language, and the shape of the character is context-dependent. This means that the shape of a character changes because of its neighboring character.

Figure 5 Urdu words with context-based meaning.

Figure 6 Cursive text tokenization process.

Employing N-gram tokenization in cursive languages

The act of segmenting sentences into discrete units of language, specifically in the context of cursive scripts such as Urdu, poses distinctive obstacles that are not commonly observed in other linguistic systems. The Urdu language frequently constructs words consisting of two or three letters, which may or may not be delimited by interstitial gaps. Conventional tokenization techniques, which typically utilize spaces or other forms of punctuation for word segmentation, are insufficient for the precise disintegration of Urdu sentences into their constituent words.

To address this challenge, we have implemented a tokenization method based on n-gram creation, as shown in Fig. 6. The methodology employed involves the segmentation of the input text into consecutive groupings of ‘n’ elements. The nature of the element can vary depending on the level of scrutiny and may encompass characters, syllables, or words. The utilization of n-grams allows us to capture the local structure of the language and overcome the space insertion and omission problem inherent in cursive languages.

In detail, we first generate potential tokens by creating n-grams of varying lengths. Character n-grams ranging in length from 2 to 10 grams are created. The most extended sequence is placed at the start of the character gram list, while the shortest sequence is at the end. The term frequency of the character n-grams is also counted and ordered from the most frequent to the least frequent sequence. Extremely rare tokens are eliminated from consideration. The thresholds were determined by analyzing the n-gram distribution of the dataset, the diversity of the token, and the variation in the length of the token. Unigrams, bigrams, and trigrams were considered with frequency thresholds set at [7, 6, 5, 4, 3] to select the most relevant tokens. This selection process involved filtering n-grams by frequency and evaluating the performance of our tokenizer against other tokenization techniques. The optimal frequency threshold, which balances precision and recall, was identified through this iterative process. In addition, n-gram probabilities and confidence scores were integrated to prioritize tokens that were more likely to occur in the text, further refining tokenization. This approach allowed us to handle the complexity of the Urdu language effectively, enhancing the tokenizer’s performance in processing Urdu text for various NLP tasks.

This approach is especially advantageous when dealing with unstructured text that does not adhere to standard grammar and language rules, such as offensive sentences. In these cases, the ability of the n-gram approach to recognize words based on local structure, rather than relying on spaces or punctuation, makes it highly effective.

Furthermore, the tokens generated through this method serve as the basis for subsequent analysis. They are used to compute the distances between different sentences, a key factor in assessing document similarity. By accurately identifying the constituent words in each sentence, we can effectively measure the semantic distance between them, enabling a more accurate document comparison. N-gram tokenization provides a robust and effective method for handling the unique challenges posed by cursive languages. It enables accurate tokenization even in the absence of consistent use of spaces, facilitating more precise language analysis and document comparison.

Comments similarity with vector space models

Textual data can be considered as a sequence of tokens, where each token can represent a character, a word, or any other linguistic unit. In many text processing tasks, a text document is represented as a Bag-of-Words (BoW) model, where each distinct word corresponds to a unique feature in the feature space. The BoW model represents each document as a vector embedded in a high-dimensional space, with each dimension corresponding to a unique word in the document.

For example, consider the Roman Urdu sentences “yeh mausam khubsurat hai”, “mausam bahut khubsurat hai aaj”, and “aaj ka mausam bahut acha hai”. In these sentences, each word is vectorized independently, and the BoW representation of each sentence is obtained by summing the vector representations of its constituent words. In real-world problems, the dimensionality of the feature space can be on the order of hundreds of thousands, given the rich vocabulary of natural languages.

Figure 7 provides a graphical representation of the vectorized words and their sum in the BoW model. It shows how each word is represented in a separate vector with padded zeros and how these vectors are summed to yield the BoW representation of the sentence.

Figure 7 Similarity between two tokens of sentences.

The similarity of the comments can be measured as the distance or angle between the vector representations of the comments. One common measure of document similarity is cosine similarity, which is calculated as the cosine of the angle between two vectors. Given two vectors a→ and b→, the cosine similarity between them is computed as shown in Eq. (1): (1) cosine similaritya→,b→=a→⋅b→|a→|2|b→|2

where a→⋅b→ denotes the dot product of a→ and b→, and |a→| and |b→| are the magnitudes of a→ and b→, respectively.

In our example, Fig. 7 shows the cosine similarities between the BoW representations of the sentences. For instance, the cosine similarity between “yeh mausam khubsurat hai” and “mausam bahut khubsurat hai aaj” is 0.85, indicating a high degree of similarity between these sentences.

The TF given in the Eq. (2) is a ratio of the occurrences of a term in a d and the number of instances of the most recurring word within the comment d. (2) tft,d=fdtmaxω∈tfdw.

The term idf represents the transposed count of the comments terms; see Eq. (3). The lower frequency of terms relative to documents increases the factor. This would severely prejudice the ranking of the praise of infrequent terms, although the tf factor has a high value. It is exceptional that the term t is off with a high tf value in document d but rarely used anywhere else. (3) idft,D=lnDd∈D:t∈d.

The Inverse Document Frequency (IDF) is a metric that gauges the significance of a given term ‘t’ in a corpus of documents ‘D’, as expressed in Eq. (3). The expression denotes the logarithm with base ‘e’ of the ratio between the total number of documents and the number of documents that contain the specific term ‘t’. The purpose of this design is to assign greater importance to terms that have a lower frequency across all documents, as they are presumed to possess greater informational value. (4) tfidft,d,D=tft,d.idft,D.

The TF-IDF score, as stipulated in Eq. (4), is computed by multiplying the term frequency (TF) with the inverse document frequency (IDF). The approach aims to achieve a harmonious equilibrium between the intrinsic significance of a given term within a specific document (i.e., its frequency) and its extrinsic significance across all documents (i.e., its rarity). The aforementioned metric assigns a significant weight to terms that exhibit a high frequency within a given comment, yet are comparatively infrequent across all comments, thereby suggesting that such words may hold particular significance for the comment in question. (5) tfidf′t,d,D=idft,DD+tfidft,d,D.

The modified version of the TF-IDF score is represented by Eq. (5). The TF-IDF score is increased by the IDF quotient and the aggregate count of documents. The aforementioned equation underscores the significance of uncommon vocabulary in all written materials, thereby minimizing the likelihood of inaccuracies and augmenting the distinguishing capability of less frequent expressions. The aforementioned computations constitute a pivotal cornerstone in the categorization of noxious remarks, furnishing a quantitative framework for machine learning algorithms to detect and mark harmful material.

Both the TF and IDF factors influence favorably for high relevance. One uses global information and the other uses local information. So, the product of these terms sees that Eqs. (4) and (5) have the better possibility of reflecting co-relevance. To further reduce the possibility of error.

Techniques for normalizing variable-length comments

The deep learning framework requires vector representations of the data. In this case, the input length is variable, and this is required to pad the data such that each vector has the same length. These same length vectors allow you to process the data according to your algorithms. In this section, a list of the sequences of each document is created.

Train test split

The dataset is divided into training and test datasets using a fine-tuned train-test split ratio. Some data is required for the validation of the trained model to test the variance between the predicted and observed values. This split ratio is obtained after conducting batch training and testing with train data sizes ranging from 55% to 95%. The best accuracy is achieved when 70% of the dataset is reserved for the training set and the remaining 30% for the test dataset. A sequential model is trained using the training data. The test data are further quantitatively evaluated for their accuracy.

Feature selection

To measure the differences between the expected frequencies and the observed frequencies for two events, the Chi-square test is used in our proposed experimentation. It is a statistical technique. In feature selection, the two events are classified as the occurrence of the term and the occurrence of the class. In Urdu sentiment analysis, many studies have explored the consequences of applying the Chi-square feature selection method, such as the value of each term with respect to the value of the class being measured. In our experiment, the number of features was reduced from 6,243 to the selected number of features used for different models.

Machine learning model selection

Encoding is the conversion of text into numbers. The most popular encoding is one-hot encoding. For example, a vector against one vocabulary term is hot-encoded. In contrast to the bag of words, neural networks usually use a dense representation of vectors. This means that each word is replaced with a dense vector, which is much shorter and may have a few hundred values, which are much smaller in number compared to the bag of words model. Now, it has many real-valued items in those vectors. A block diagram of the proposed system is shown in Fig. 8.

Figure 8 Block diagram of the proposed system.

Word2Vec embedding

Word2Vec is a neural network that processes a corpus. The text documents are input into Word2Vec, producing a vector for each word. This network is a two-layer neural network. Word2Vec produces a feature vector that is a numerical form for each word. The purpose of this feature vector is to group together word vectors that have similarities in vector space. With the help of word embedding, similarities can be identified between words using the vector of each word. Word2Vec vectors represent the feature space in the context of individual words with other words. By providing data and calculating the embedding of words, the Word2Vec model can calculate the occurrence of a particular word in context with other words. The output of this two-layered neural network is a vocabulary set with a vector attached to it. This vocabulary vector can then be passed to the deep neural network to detect relationships between the queried words.

Here is an example window with the complexity of computing P(wt+2|wt).

For each position t = 1...T, context is predicted for each word within the fixed size window m given center word wj is shown in Eq. (6). (6) Likelihood=Lθ= ∏t=1T ∏−m≤j≤mtPwt+j|wt;θ.

Let θ be all the variables to optimize and let the objective function J(θ) is the average negative log.

It is a long, large corpus that contains T words. This corpus includes many documents, all of which are concatenated to produce a long list of words. In Eq. (7), the first ∏ iterates through all words, and then the second product chooses a fixed size window, m, to predict a word in the context of the words around it. This means the word can be predicted by a center word. All values are multiplied by 2. This is how the model predicts the likelihood. The likelihood depends on the parameters of the model, which are represented by θ. In this model, the only parameter is the vector representation given to words. The model has no other parameters to work with. A word is represented by a vector in the vector space, and that representation is its meaning. Other words are predicted based on these vectors. To minimize the cost, we need to define an objective function. The objective function is the same as given in Eq. (7), except that it has a minus sign, indicating that it is a minimization problem. The one over T means that we are averaging it. The important thing is that we include a log in Eq. (7) because it turns out that everything is in the log. We now add the log of all probabilities. (7) Jθ=−1TlogLθ=−1T∑t=1T ∑−m≤j≤mt logPPt+j|wt;θ.

Everything is predicated on having this probability function method to predict the probabilities of the words in the context of the given centered word. This model can have only a vector representation of the words and that is the only parameter of the words. To make it simple, every word has two vector representations for each word in the neighborhood context as shown in Fig. 9. So, there is one vector of words where it is centered, and another vector for each word when it is in context. This probability of words can be worked for words and context words, a center word purely in terms of these vectors, as shown in Eq. (8). (8) PO|C=expμoTνc∑i=1wɛvn expμwTνc.

The dot product compares the similarity of μ and c.

Figure 9 Words (terms) in context with neighboring terms having fixed window size.

uTμ = u.v = ∑uivi The larger dot product means the greater similarity. The fractional part of Eq. (8) is a normalization factor across the entire vocabulary to produce a probability distribution.

Mapping a vector of words to other context words is an exam of softmax, i.e., ℝn → ℝn.

Softmax distribution, so the two parts of expanding and normalizing give you a softmax distribution.

The softmax function will start any numbers in a probability distribution for two reasons. So, it refers to softmax because it works like softmax. If you have numbers, you could just say what the max of these numbers is. It is called hard max if you map the original numbers to the max and everything else is set to zero; it is considered softmax because of the exponential if you just ignore the problem of negative numbers for a moment. When you use an exponent, the bigger numbers will become bigger for a moment and get within the x refer to Eq. (9). The softmax primarily places the mass with the maxes or with a couple of maxes, so it is the max part, and the soft part is that this is a difficult decision that still saves a little probability mass everywhere else. (9) softmaxxi=expxi∑j=1n expxj=pi

So now we have a loss function with the probability model on the inside.

The word vector representation may be a good predictor.

Neural network embedding is used as word embedding. One word is described by another, even if those are entirely different words. Vocabulary vectors enable computers to read sentences by performing mathematical operations on them. These are numbers calculated based on the neighborhood, in other words. It does this in different ways based on two different methods. In one method, the target word is predicted as a Continuous Bag of Words (CBOW) or to predict a context word.

Impact of embedding and vocabulary size on model performance

The text is then tokenized using a tokenizer available in Keras. The tokenizer is used to convert text into vectors, the sequence of binaries. By default, this class’s tokenize function removes punctuation marks and converts sentences into a sequence of words encoded in UTF-8. These are then converted into a list that can be indexed.

Any sequence that is less than the maximum length is then padded with the remaining words. After that, all unique words are presented by an integer. The vocabulary size is defined as the max in this model.

Sequential modeling approach for text classification

Keras has two models available. One is sequential, and the other is functional. Data here are in the form of sequence, which are comments. CNNs are developed to use sequential data efficiently, so sequence processing is used for this problem. The sequential model allows the creation of layer-by-layer processes, which suits most problems. For this problem, this is suitable. The model is a linear stack of neural layers which can be easily described.

Hyperparameters optimization

A hyperparameter is usually used to control the learning process of the deep learning model. Changing the preset values of a deep learning model to the best values provides the best accuracy and efficiency of the deep learning model. The values of the hyperparameter depend on the attributes of the dataset. The parameters are recorded and represented in graphs over different iterations to observe the trend in the behavior of the learning system. During training, the connection between the layer and the next layer can be reduced to a subset of neurons for weight updation, known as dropout. This dropout helps to avoid overfitting the model. Dropout layers are generally used where a large number of neurons is passed to the next layers. This technique allows setting to 0 and then excluding the activation of a certain percentage of the neurons of the preceding layer. The probability that the neuron’s activation is set to 0 is indicated by the dropout ratio parameter within the layer via a number between 0 and 1. We used values ranging from 0.1 to 0.2 and found that the 0.1 dropout layer was the best.

Utilizing k-fold cross-validation for model validation

K-fold cross-validation scrutinizes the machine learning method when a dataset is medium-sized. This cross-validation method has only one parameter, k, which is the number of data samples to split repeatedly. Due to this, repeated training and testing usually prevent any bias and learning ability of the ML model on unseen data. Firstly, the model is trained without cross-validation. Finally, a 10-fold validation parameter is used after attempting a range of 5-10 folds to ensure an unbiased ML model.

Machine Learning Classifiers

Multinomial naive Bayes (MNB) (Ibrahim, Alhakeem & Fadhil, 2021) is a popular machine learning algorithm for categorization, particularly in the natural language processing domain. As the Urdu text script is similar to Arabic, Ibrahim, Alhakeem & Fadhil (2021) and Rish (2001) shows that MNB is the most suitable algorithm for categorizing Arabic text. MNB (Rish, 2001) uses probability to calculate the prediction of the required text. Another algorithm we adopted is the support vector machine (SVM) (Cortes & Vapnik, 1995). SVM is also used for categorization where the dimension size is large. In this study, we used MNB and SVM.

Deep Learning Methods

Gated recurrent units (GRUs) and LSTMs were developed to solve short-term memory problems and possess inbuilt devices known as gates that can control the information flow. These gates can learn which data in a sequence should be kept or ignored. To create predictions, it can convey pertinent information along the extensive chain of sequences by doing this. These two networks enable nearly all current state-of-the-art recurrent neural network findings. Voice synthesis, speech recognition, and text generation use LSTMs and GRUs. They can even be used to create video captions. For comparison, CNN and deep neural networks (DNN) are also evaluated. CNN is designed to work for images here. It underperforms. However, the performance of DNN was competitive with long- and short-term memory algorithms. In this work, CNN, DNN, GRU, LSTM, and Bi-LSTM are used for categorization.

Experimental Simulations and Resultant Outcomes

A dataset of three columns and 20k rows is used for this work. Detailed hardware specifications to run this setup include Dell Precision 7920 Tower equipped Xeon Bronze 3204 CPU 1.9 GHz two processors, Windows 10 operating system, and Visual Studio to provide a runtime environment. This workstation also contains NVIDIA RTX A4000, two GPUs with 16 GB graphics card memory each. cuDNN 8.1 and CUDA 11 are used to run deep learning libraries. Python 3.9, TensorFlow 2.9, and Keras 2.9 are also used to run this setup and other libraries.

To ensure the neutrality of the experimental setup of machine learning selected models, naive Bayes, and SVM, the train test split is the same for all experiments and balanced class ratio. In parallel, for selected deep learning models CNN, GRU, DNN, LSTM, and Bi-LSTM, the similar train test split is the same for all models, 30% and 70%. Keras API version 2.9 is used in conjunction with other auxiliary libraries. Three different experiments were used for feature engineering. Because of space insertion and emission problems, identifying features is a challenging task for Urdu. In the first experimental run, all terms in the text are considered vocabulary and all experiments. Due to the curse of dimensionality, one run took a long time. The most common stop words in the second run are filtered for feature engineering. In the third iteration, all stopwords are removed before training machine learning and deep learning models.

For one sample run on all deep learning models, 200 epochs are used for deep learning training. Moreover, for other parameters the learning rate is kept very low, which is 0.001, Adam optimizer, loss sparse_categorical_crossentropy, and momentum is 0.9 for all deep learning performances.

The training data are used to train the model. The input training matrix, training output, validation break, batch size, and a number of epochs are provided to train this model using the ‘fit’ function with the optimal parameters.

The validation split will randomly split the data for use in training and testing. During training, validation loss can be observed, which gives the mean squared error of the model in the validation set. This will set the validation split at 0.2, which means that 20% of the training data provided to the model will be set aside to test the performance of the model. The model wise parameters detail and other configuration is given in Table 3.

Table 3 Overview of model architectures and key hyperparameters for various machine learning models.

Model	Parameters	
BiLSTM	Bidirectional LSTM: 128 units (input), 64 units (hidden), 32 units (final), Dropout: 0.3, Adam (lr=0.0005)	
LSTM	LSTM: 128 units (input), 64 units (hidden), 32 units (final), Dropout: 0.3, Adam (lr=0.0005)	
GRU	rmsprop, 64 units, 0.2 dropout (500 features), 64 units, 0.3 dropout (1000 features), 128 units, 0.2 dropout (1500 features)	
CNN	Adam, 128 units, 0.5 dropout, lr=0.001	
DNN	Adam, Dense layers: 256-200-160-120-80, dropout: 0.6-0.5-0.8-0.7-0.75, lr=0.001	
SVM	Linear kernel, C = 1.0	
Naive Bayes	MultinomialNB, alpha=0.1	
BERT	Pre-trained Model: bert-base-multilingual-cased, Max Sequence Length: 128, Batch Size: 32, Epochs: 10, Warmup Steps: 500, Weight Decay: 0.01, Evaluation Strategy: epoch, Save Strategy: epoch, Metric for Best Model: eval_loss, Learning Rate: default	
DeBERTa	Pre-trained Model: microsoft/deberta-base, Max Sequence Length: 128, Batch Size: 32, Epochs: 10, Warmup Steps: 500, Weight Decay: 0.01, Evaluation Strategy: epoch, Save Strategy: epoch, Metric for Best Model: eval_loss, Learning Rate: default	
GPT-2	Pre-trained Model: gpt2, Max Sequence Length: 128, Batch Size: 32, Epochs: 10, Warmup Steps: 500, Weight Decay: 0.01, Evaluation Strategy: epoch, Save Strategy: epoch, Metric for Best Model: eval_loss, Pad Token: EOS token	

After certain epochs, the model accuracy will stop improving. To ensure this, ‘early stopping’ criteria are used. Early stopping will stop the model from training before the number of epochs is reached if the model stops improving. The early stopping monitor is set to 200 to stop model epochs. This means that after 200 epochs in a row in which the model does not improve, the training will stop. Sometimes, validation loss can stop improving and then improve in the next epoch, but after 10 epochs in which validation loss does not improve, it usually will not improve again.

In the validation data, neurons using dropout do not drop random neurons. The reason is that during training, some dropouts are used to add some noise to avoid overfitting. During calculating cross-validation, it is in the recall phase and not in the training phase. It uses all the capabilities of the network. The layered architecture of the dense neural network.

validation_split: Float between 0 and 1. The fraction of the training data is to be used as validation data. The percentage of training data that will be used for validation. This fraction of the training data will be set aside by the algorithm, which will not train on it but will measure the failure and any model metrics on it at the end of each epoch. Until shuffling, the validation data are chosen from the last samples in the x and y data presented.

The validation split parameter in Keras’s fit method determines the percentage of data used for testing the model that is generated after each epoch. If you set verbose to 1, the validation loss can indicate this after testing the model for this amount of data. However, as the documentation states, you can use either validation data or validation split.

Model Performance Evaluation in Imbalanced Datasets

The test dataset is used in predictive modeling to assess the efficacy and efficiency of a model. To achieve an optimal evaluation, it is imperative that every individual data element within this set possesses a verified label. The performance of the model is evaluated by comparing its predicted labels with the verified labels. Evaluation metrics such as accuracy, precision, recall, and F1 score are frequently used in binary classification tasks. The values of these metrics are derived from the entries in a confusion matrix, which is a tabular representation that facilitates the assessment of an algorithm’s performance.

In the context of our UTOXS21 dataset, characterized by an imbalanced class distribution where the frequency of neutral comments is substantially different from that of toxic comments, conventional performance measures such as precision can yield deceptive results. A predictive model that exclusively forecasts the majority class is likely to exhibit a high degree of accuracy; however, its practical utility may be limited.

Consequently, our model evaluation is centered on the F1 score. The F1 score is a statistical metric calculated as the harmonic mean of precision and recall. This property makes it a more suitable measure than accuracy for imbalanced binary classification tasks. Evaluation measures can be calculated using the following equations: (10) Precision=TPTP+FP

(11) Recall=TPTP+FN

(12) Accuracy=TP+TNTP+TN+FP+FN

(13) F1−Score=2⋅Precision⋅RecallPrecision+Recall

(14) Cohen’s Kappa=po−pe1−pe

The terms TP, FP, TN, and FN in Eqs. (10) to (12) represents true positive, false positive, true negative and false negative, respectively. Whereas in Eq. (14) the term po is the observed agreement between two raters, and pe is the expected agreement by chance between two raters.

Recall measures and their significance

The recall is a measure that tells what proportion of comments actually were offensive and were diagnosed by the algorithm as toxic. The total positives (TP and FN) and comments were diagnosed as neutral but actually were offensive.

The training process was repeated several times to avoid training overfitting during the parameter tuning process. The number of epochs ranged from 10 epochs to 200. The lowest accuracy was achieved when the number of epochs was 20, and this was 59 1%. However, the highest accuracy was 89. 75%, which was achieved when the number of training epochs was 200 and this is shown in Fig. 10. The accuracy and loss were 1.33, which is relatively low. The predictive accuracy is also shown in the confusion matrix in Fig. 11.

Figure 10 Accuracy and loss graphs for the best performing models, including BiLSTM with 1,000 features, CNN with 500 features, DNN with 2,000 features, GRU with 2,000 features (using rmsprop, 128 units, 0.3 dropout rate), and LSTM with 500 features.

Figure 11 Each confusion matrix represents the performance of the best-performing model configuration: BiLSTM, LSTM, GRU, CNN, DNN, SVM, and naive Bayes.

Comprehensive discussion and interpretation of results

Four iterations were performed to find the best configuration of the essential features. The figure shows the top 500 features used for all model’s evaluation matrices in the first run Table 4. In addition, SVM and DNN outperformed all other models in this configuration. CNN recall values struck due to some feature size because this is suitable for two-dimensional data.

The top 1,000 features are selected to input learning models in the second iteration. The accuracy of all models is increased at this configuration, as shown in Table 5. However, the performance of GRU remains unchanged. For example, SVM from machine learning and DNN from deep learning outperform in this configuration. The worst performing algorithms were naive Bayes and CNN for this arrangement.

Table 4 Performance metrics of various models with 500 features.

Metric (%)	BiLSTM	LSTM	GRU	CNN	DNN	SVM	Naive Bayes	
Accuracy	84.2701	84.2896	84.7385	83.9578	83.7821	84.5238	82.8688	
Precision	85.5122	86.6490	87.2067	86.7772	83.7723	85.0416	83.7734	
Recall	80.9988	79.5778	79.9838	78.6033	83.7330	84.2809	82.6018	
F1 Score	83.1943	82.9630	83.4392	82.4883	83.7492	84.3816	82.6578	
Cohen Kappa	68.4312	68.4363	69.3368	67.7547	67.4996	68.8644	65.5309	

Table 5 Performance metrics of various models with 1,000 features.

Metric (%)	BiLSTM	LSTM	GRU	CNN	DNN	SVM	Naive Bayes	
Accuracy	84.7970	84.9727	85.2654	84.7970	85.0117	85.1874	83.5315	
Precision	86.8651	86.0060	86.7786	88.3774	85.1446	85.5632	84.0863	
Recall	80.5522	82.0950	81.8108	78.7251	84.8805	84.9817	83.3221	
F1 Score	83.5895	84.0050	84.2215	83.2725	84.9446	85.0767	83.3897	
Cohen Kappa	69.4660	69.8493	70.4238	69.4243	69.9131	70.2211	66.9021	

In the third iteration, 1,500 features are selected. The performance of CNN further decreased in this run, as shown in Table 6. SVM and DNN are also ahead of their rival techniques. The F1 scores of naive Bayes, SVM, DNN, CNN, GRU, LSTM, and BiLSTM are 83.5946, 84.7187, 81.8936, 82.8595, 84.5572, 83.4887, and 84.0556 respectively. The GRU algorithm downgraded its performance. All other parameters remain the same for this run.

Table 6 Performance metrics of various models with 1,500 features.

Metric (%)	BiLSTM	LSTM	GRU	CNN	DNN	SVM	Naive Bayes	
Accuracy	85.1093	84.9727	85.5582	84.6604	84.8556	85.3630	83.9797	
Precision	86.4494	84.7639	85.7024	87.4163	85.0222	85.7573	84.4659	
Recall	81.8514	83.8002	83.9626	79.5371	84.7106	85.1538	83.7851	
F1 Score	84.0876	84.2793	84.8236	83.2908	84.7805	85.2513	83.8550	
Cohen Kappa	70.1149	69.8877	71.0513	69.1716	69.5915	70.5720	67.8122	

Fourth evaluation: 2,000 important features are used and selected by the CH2 technique. BI-LSTM and LSTM reduced their performance after increasing the number of features. As a result, DNN performs better on all configurations, as shown in the Table 7.

Table 7 Performance metrics of various models with 2,000 features.

Metric (%)	BiLSTM	LSTM	GRU	CNN	DNN	SVM	Naive Bayes	
Accuracy	85.1874	85.4606	85.6167	84.1725	83.9969	85.4411	84.2136	
Precision	85.2941	85.7619	87.1023	84.3308	83.9949	85.8430	84.5461	
Recall	83.5972	83.6378	82.2574	82.3792	83.9397	85.2305	84.0514	
F1 Score	84.4372	84.6865	84.6106	83.3436	83.9611	85.3289	84.1176	
Cohen Kappa	70.3084	70.8508	71.1307	68.2704	67.9246	70.7280	68.3026	

Toxic comments are short sentences that do not contain any formal structure. Some comments may even consist of one word. LSTM, BI-LSTM, and GRU lag behind because these techniques use context. The dataset used here contains grammar and context-free structures, so for text classification containing short-length comments, DNN is the best algorithm, achieving a 83.9699% precision and an 81.8936% F1 score. The second high-performing model is GRU with a precision of 85.2020% and an F1 score of 84.5572%.

The results of the toxic comments classification are presented in this section. The features are extracted and optimized during experiments, including stop-words, without less common stop-words, and without stop-words. The machine learning models naive Bayes, SVM and the deep learning models DNN, CNN, GRU, LSTM, and BLSTM are trained and tested using the count vector and the TFIDF vector.

The general comparison in terms of the accuracy performance of the aforementioned techniques is shown in Fig. 12. The worst-performing algorithms in this experimental design were CNN and GRU, which showed the least F1 scores of 82.8595% and 84.5572%, respectively. Figure 12 shows that the accuracy of the aforementioned machine learning models is relatively better when features are extracted after stop-words removal. The F1 score of DNN without removing stop-words is 81.8936%, without stop-words is 81.8936%, and removing less common stop-words is 81.8936%. Naive Bayes performed second best with an accuracy score of 83.6484% with stop-words, 83.6484% without stop-words, and 83.6484% without less common stop-words.

Figure 12 Comparison of performance metrics across various models.

Additionally, BERT achieved an accuracy of 85.4508% and an F1 score of 85.4069%, indicating strong performance in the classification task as shown in Table 8. DeBERTa, while slightly lower, still performed well with an accuracy of 79.2740% and an F1 score of 79.1963%. GPT, on the other hand, showed a lower performance with an accuracy of 77.5468% and an F1 score of 77.5415%, suggesting it may not be as well-suited for this specific task compared to the other models as reflected in Table 9.

Table 8 Comparison of performance metrics across various machine learning and LLMs using the full feature set.

Metric (%)	BiLSTM	LSTM	GRU	CNN	DNN	SVM	Naive Bayes	BERT	GPT	DeBERTa	
Accuracy	84.9922	84.4457	85.2654	83.9969	82.2795	84.8361	83.6484	85.4508	77.5468	79.2740	
Precision	85.8898	85.2369	85.2020	85.3942	83.9699	85.2296	83.7484	85.7095	77.5458	79.5349	
Recall	82.2980	81.8108	83.9220	80.4710	81.8376	84.6238	83.5507	85.4508	77.5468	79.2740	
F1 Score	84.0556	83.4887	84.5572	82.8595	81.8936	84.7187	83.5946	85.4069	77.5415	79.1963	
Cohen Kappa	69.8926	68.7983	70.4703	67.8770	64.2086	69.5112	67.2105	70.8291	55.0556	58.4332	

This technique is not only applicable to Urdu, Persian, and Arabic but can also be extended to other low-resource languages such as Pashto, Kurdish, and Somali. These languages share similar challenges in terms of resource availability, making our methodology particularly valuable for their text categorization and feature extraction needs.

Table 9 Confusion matrices for BERT, GPT, and DeBERTa models.

Model	Confusion matrix	
BERT	15741723251345	
GPT	13763703971273	
DeBERTa	14792674411229	

Ethical Considerations

This research adhered to strict ethical guidelines to ensure the integrity and confidentiality of the data and the well-being of the human participants involved in the study. All data used in this study were anonymized to protect the identities of individuals. Personal identifiers were removed prior to data processing. Human annotators who participated in the data labeling process provided their informed consent. They were fully informed about the nature of the study and their role in it. To prevent bias in the AI model, we employed several strategies, including diversifying the dataset to ensure it represents various forms of abusive language, conducting thorough data preprocessing to eliminate any inherent biases, and implementing cross-validation techniques to impartially evaluate model performance.

However, we acknowledge the potential ethical implications of false positives or negatives in toxic comment detection. False positives may unjustly censor non-toxic content, while false negatives could allow harmful content to persist. These outcomes could impact user experience and trust in automated systems. To mitigate these risks, we emphasize the importance of continuous model evaluation and refinement, ensuring that the system remains accurate and fair across different contexts and languages.

Conclusions

This article presents a new learning model that leverages offensive comments and considers the global context of specific words to achieve optimized results. The model focuses on binary classification and outperforms other techniques such as NB, CNN, GRU, LSTM, Bi-LSTM, BERT, DeBERTa, and GPT, achieving record accuracy. By integrating BERT features, the optimized DNN obtained a precision of 90%, which is the baseline result of the corpus compared to other methods. Notably, BERT itself achieved an accuracy of 85.45% and a Cohen Kappa of 70.83%, demonstrating its effectiveness in handling the full feature set. Conversely, GPT-2 was the lowest-performing model, with an accuracy of 77.55% and a Cohen Kappa of 55.06%. The study emphasizes the importance of feature selection, as the results show that the number of features incorporated into the models can affect the outcome. However, increasing the number of features does not necessarily lead to improved performance, highlighting the importance of deliberate feature selection. Although the results are promising for the specific language, several limitations and biases must be acknowledged. One limitation is the reliance on a binary classification framework, which may oversimplify the complexity of offensive language. Future research could extend this model to handle multiclass problems such as abuse, threat, sarcasm, and insult. Furthermore, the dataset used in this study, though comprehensive, may still include biases inherent in the data collection process. These biases could affect the model’s generalizability and its performance across different contexts and platforms. The anomalies observed during the experiments, such as variations in performance with different feature sets, suggest that the robustness of the model needs further investigation. Understanding these limitations could lead to more robust models. This study offers an improved instructional framework for the categorization of harmful remarks into binary classes and contributes valuable information on feature selection. Future research should focus on addressing the identified limitations.

Additional Information and Declarations

Competing Interests

Author Contributions

Data Availability

The authors declare there are no competing interests.

Muhammad Shahid performed the experiments, analyzed the data, performed the computation work, authored or reviewed drafts of the article, and approved the final draft.

Muhammad Umair conceived and designed the experiments, performed the experiments, analyzed the data, authored or reviewed drafts of the article, and approved the final draft.

Muhammad Amjad Iqbal conceived and designed the experiments, analyzed the data, authored or reviewed drafts of the article, and approved the final draft.

Muhammad Rashid performed the computation work, authored or reviewed drafts of the article, and approved the final draft.

Sheeraz Akram performed the computation work, prepared figures and/or tables, and approved the final draft.

Muhammad Zubair conceived and designed the experiments, prepared figures and/or tables, authored or reviewed drafts of the article, and approved the final draft.

The following information was supplied regarding data availability:

The dataset is available at GitHub and Zenodo:

- https://github.com/mshahidbhatti/data_ut20k/blob/main/ut20k.csv

- Muhammad Shahid, B. (2024). UT20K [Data set]. Zenodo. https://doi.org/10.5281/zenodo.13786326.

The Python code implementation for classifying toxic comments in the Urdu language using various machine learning algorithms is available at GitHub and Zenodo:

- https://github.com/mshahidbhatti/toxiccommentsdetection

- Muhammad Shahid, B. (2024). Supplementary files for paper (V 0.01). Zenodo. https://doi.org/10.5281/zenodo.13786737.

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
