# Peer review of "Leveraging deep learning for toxic comment detection in cursive languages"

_PeerJ Computer Science, doi:10.7717/peerj-cs.2486_

## Round 0.1 · original submission · Major Revisions

Dear authors,

Thank you for submitting your article. Feedback from the reviewers is now available. It is not recommended that your article be published in its current format. However, we strongly recommend that you address the issues raised by the reviewers, especially those related to readability, experimental design and validity, and resubmit your paper after making the necessary changes. Before submitting the paper following should also be addressed:

1. Please write research gap and the motivation of the study. Evaluate how your study is different from others. More recent literature should be examined.
2. The values for the parameters of the algorithms selected for comparison are not given.
3. The paper lacks the running environment, including software and hardware. The analysis and configurations of experiments should be presented in detail for reproducibility. It is convenient for other researchers to redo your experiments and this makes your work easy acceptance. A table with parameter settings for experimental results and analysis should be included in order to clearly describe them.
4. Equations should be used with equation number. Please do not use “following”, “as follows”, etc. Explanation of the equations should be checked. All variables should be written in italic as in the equations. Their definitions and boundaries should be explained. Relevant references should be given for the equations.
5. You should include in-text citations of tables and figures in the text so that the reader knows which part of your written work refers to the tables.

Best wishes,

·

Basic reporting

1. The article's language is generally good and mostly clear. However, there are several grammatical errors and awkward phrasings that could hinder understanding for a the readers. For example, "This study proposes a novel model to identify salient features from Urdu sentences using transformers to identify and flag the toxic comments using deep learning binary classification of the text." This sentence contain repetitive phrases, which makes it hard to follow. The last two paragraph of "Background" section are also repetitive. Consider revising sentences that are overly complex or improperly structured to improve clarity and readability. Please do not fully entrust to chatGPT to check and rewrite the writing.
2. The overall article provides a robust review of existing literature and contextual background, particularly in the area of toxic comment detection across various languages, including low-resource languages like Urdu. References are well-cited, demonstrating a solid grounding in the relevant research.
3. There is inconsistencies on the explanation of the data. In the subsection 2.2, it stated that the dataset contains around 43% classified as neutral and 57% comments identified as toxic. However, in table 2 the dataset has 10.000 comments as toxic and 10.000 comments as neutral.
4. On the results part, the author always mention "accuracy", however on the table of results accuracy was not evaluated. I also notice that the results is quite questionable. GRU and LSTM should have similar results since both of them are the variant of RNN. I would suggest to double check the configuration of GRU models. I also have a concern with the CNN results which is very low. With that results we can see that the models did not properly learn the problem. To have a better interpretation of the results, please check again the experiment.

Experimental design

1. The article exploited several machine learning models for detecting toxic comments in Urdu, aligning with the journal's focus on computational methods and innovative applications in computer science. The topic is highly relevant given the increasing importance of online communication moderation.
2. The research question is mainly focusing on Urdu, a low-resource language. The paper states that existing tools are insufficient for this language, which underscores the relevance and necessity of the research. However, I can easily found several related works which also focused on abusive language detection in Urdu. Please check Akhter et al., 2020, Haq et al., 2020, and Humayoun, 2022. I notice that building dataset and effort to build machine learning models for detecting abusive language in Urdu (where author said low resource language) is not a new research objective. However, a new resource is also always beneficial.
3. To fully meet the journal standards, the authors should include a section detailing the ethical guidelines followed during the research, such as anonymization of data and informed consent (if human annotators were involved), and any measures taken to prevent bias in the AI model.
4. The methods section provides an overview of the data collection, model training, and evaluation processes. Specific details on the architecture of the neural networks used and the parameters selected for training are given, which is good for replication purposes.

Validity of the findings

1. The manuscript could further explain the impact of this work by discussing potential applications of the model beyond the academic sphere, such as in social media moderation tools or online community management platforms. Additionally, the paper should encourage replication by detailing how the model might be adapted or tested with other low-resource languages, thus broadening its relevance and utility.
2. While the conclusions are well-articulated, they could be improved by providing a deeper analysis of the results in the context of existing research. Discussing any limitations of the study, potential biases in the model, or anomalies found during the experiment could provide a more balanced view and suggest areas for further research.

Additional comments

While the paper describes the model's architecture and some aspects of its implementation, it does not provide enough detail about the statistical methods and the experimental setup, such as hyperparameter tuning, model validation processes, and error analysis. This lack of methodological transparency can make it difficult for other researchers to replicate the study or verify the claimed improvements over existing models. This results presented in this paper also need to be further validated to provide better analysis.

·

Basic reporting

There are several technical writing issues with the manuscript. A few are given below:
In 0.1 section, authors claims that their “study presents several significant advancements and novel approaches”. The word ‘several’ is not suitable according to the context of the sentence. Suggestion is to use ‘few’ instead of 'several'.
Need to merge the paragraphs of less than 3 lines like line 181 and 182 with above or below paragraph. Same for 310 and 311 lines. Also 330, 331 and 332, 333.
I would suggest reviewing the manuscript carefully and correcting all the issues.

Experimental design

The manuscript is novel and has broader scope in the scientific era as the use of cursive language is being increasing day by day. Gaps were identified from the literature and research questions have been addressed. Methdology is explained briefly.

Validity of the findings

Manuscript is novel and has a larger impact on society where cursive languages have been used. All the dataset have been described with facts and figures. I would suggest making this dataset public by including its link in the manuscript as a footnote in the dataset section. This is also useful to reproduce the results.

Additional comments

On line 323, ‘toxic and non-toxic files are nearly equal in numbers’. Is it not 57% and 43% ratio? As it is mentioned in the 2.3 section, dataset is imbalanced and the use of ‘nearly equal’ is inappropriate to use here. And this information is not aligned with Table 2 information where both classes have equal number of comments.
On line 353 and in Figure 4, “N-grams that meet a certain frequency and length threshold are considered as valid tokens.” What was that frequency and length threshold? How did authors choose this frequency and threshold values?
Figure 5 needs more explanation for better understanding. Which two sentences are represented here and what are the nodes and edges and the labels?
In section 3.3, because this manuscript is about comments classification, use of ‘document’ word is inappropriate in the title and the text of section 3.3. It is recommended to replace ‘document’ with ‘comment’
On 396 line, ‘shown in Fig. ??” Figure number is missing.
Is the split ratio 85-15% used for both machine learning and deep learning models?

In section 3.6, from 20k comments, only 950 were the total features finally used for experiments? And these features were n-grams of some length?

---

## Round 0.2 · Minor Revisions

Dear authors,

Thank you for addressing the reviewers' concerns. One of the two reviewers did not accept to review of the revised manuscript. The other reviewer thinks that your manuscript requires some minor revisions. We do encourage you to address the concerns and criticisms of this reviewer and resubmit your article once you have updated it accordingly.

Best wishes,

·

Basic reporting

- The revised manuscript shows improvements in language clarity and structure. Consider a final proofread focusing on simplifying complex sentences and further reducing repetition to enhance clarity and scientific tone. Consider also to use professional language editing service.
- Ensure that all data related to the study, including raw data or a sample thereof, is available or clearly linked for verification and replication, adhering to the journal’s data transparency policy.

Experimental design

- The authors should continue to highlight specific innovative aspects of their methodology, particularly how it might be applied to other similar low-resource languages.
- Please discuss any potential ethical implications not already covered, such as the impact of false positives or negatives in toxic comment detection.
- Consider adding specific version for software or library and detailed parameter settings for all models to ensure that other researchers can precisely replicate the study.

Validity of the findings

- Further explore and discuss the real-world implications of deploying such models, such as in social media platforms, and the potential challenges in practical implementations.
- It would be beneficial to include a comparison of results with state-of-the-art models in high-resource languages to provide a benchmark for evaluating the performance of the proposed model.

Additional comments

- The manuscript has seen notable improvements in addressing the concerns raised in the initial review. - The authors have effectively enhanced the clarity, depth, and ethical considerations of their work.
- Minor improvements in language and further detail in methodological transparency could still enhance the manuscript.

---

## Round 0.3 · Minor Revisions

Dear authors,

Thank you for the revision. One reviewer accept the paper for publication. However, according to one of the reviewers, your paper still needs a revision and we encourage you to address the concerns and criticisms of Reviewer 2 and resubmit your article once you have updated it accordingly.

Best wishes,

·

Basic reporting

Current version has been improved.

Experimental design

Current version has been improved.

Validity of the findings

Current version has been improved.

Additional comments

Current version has been improved. I would recommend for publication on this version.

·

Basic reporting

The structure of the article is professional. Raw data is shared. some ambiguities are present in data collection, annotation, and preprocessing of the data

Experimental design

Data collection is the major contribution of this study. authors used conventional methods, and no significant method was designed or proposed in this study.

Validity of the findings

Results of the experiments are not very encouraging and almost the same with minor differences.
It would be good if they used some advanced methods of Transformer or LLMs instead of using conventional methods of deep learning and machine learning.
Authors claim to use BERT in the contribution of the study but I have not found anything in the experiments, results, and methodology.

Additional comments

Revise the abstract section. 80% of the text in the abstract is general. authors have not even included the method name (BERT), or its advantage over others. authors used 05 performance measures but only precision is given in the abstract.
Figure 1 should be a Table 1. It is in rows and columns. The table is more suitable than the figure.
For the classification of offensive comments, a reference is required in the context of Urdu text.
In contribution, why author use only BERT for toxic text classification? There are many other transformers-based BERT models and tokenizers like DistilBert, Roberta, Albert, and XLNet.
A summary of the current state-of-the-art studies in the form of a Table at the end of Section 1 would be great beneficial for the readers to get deep insight in this domain.
On line 247 for data collection, why not toxic or non-toxic comments instead of offensive or non-offensive?
In Figure 3, and on line 273, the authors normalize the text by converting all the characters two lower-case for uniformity. I am surprised to know it. A well-known paper on Urdu text classification and others claim that there are no lower-case or capital words, unlike English language words. Plz check "M. P. Akhter, Z. Jiangbin, I. R. Naqvi, M. Abdelmajeed, A. Mehmood and M. T. Sadiq, "Document-Level Text Classification Using Single-Layer Multisize Filters Convolutional Neural Network," in IEEE Access, vol. 8, pp. 42689-42707, 2020, doi: 10.1109/ACCESS.2020.2976744."

on 276 lines, tags removals, authors mentioned that collected data from Google forms contain HTML tags and tags. The authors discussed in Figure 3 and in Section 2.1 the procedure to collect the data by Google Forms. If it is true, then how the tags of HTML or others were included in the comments from the Google forms?
on line 282, special characters were converted to numeric values and then replaced with their corresponding HTML code or tags. What happened after this step? Were these tags removed in the previous step on the 276 lines? or keep it in the final text?
Lines from 290 t0 299 are about dataset description and should be added in the 2.3 section and removed from here to avoid duplication of contents.
on lines 351 and 352, the effects of n-grams with various frequency threshold distributions on the dataset must be discussed in the form of a table or a graph.
in Section 3.5, the text on lines 436-437 is very confusing and may not be accurate. It needs more details.
In Figure 8, in preprocessing, the author mentioned that they removed stopwords and stemmed the text. However, this is not included in the data collection and preprocessing sections above. Which stemmer or lemmitizer they used and which stopwords or list of stopwords were removed?
All the results discussed from Table 3 to Table 5, and Table 6, show that the performance of all the models using various performance measures is almost similar. I suggest to extend these results to one or two digits after the floating point. like in Table 6, all the models have 85% accuracy which is the same for all the models. replacing it with 85.1 or 85.8 would make a significant difference and help in good comparison of performance.

---

## Round 0.4 · accepted · Accept

Dear author,

Thank you for your hard work on the revision. One reviewer did not respond to the invitation for the revised paper. However, I see that your paper is improved and seems ready for publication after the second revision.

Best wishes,